# B-Cell Activating Factor Enhances Hepatocyte-Driven Angiogenesis via B-Cell CLL/Lymphoma 10/Nuclear Factor-KappaB Signaling during Liver Regeneration

**DOI:** 10.3390/ijms20205022

**Published:** 2019-10-10

**Authors:** Chia-Hung Chou, Cheng-Maw Ho, Shou-Lun Lai, Chiung-Nien Chen, Yao-Ming Wu, Chia-Tung Shun, Wen-Fen Wen, Hong-Shiee Lai

**Affiliations:** 1Department of Surgery, National Taiwan University Hospital and National Taiwan University College of Medicine, Taipei 10002, Taiwan; ch640124@gmail.com (C.-H.C.); miningho@ntu.edu.tw (C.-M.H.); Sauron_lai@hotmail.com (S.-L.L.); cnchen@ntu.edu.tw (C.-N.C.); wyaoming@gmail.com (Y.-M.W.); 2Departments of Forensic Medicine and Pathology, College of Medicine, National Taiwan University, Taipei 10617, Taiwan; ctshun@ntu.edu.tw; 3Departments of Pathology, College of Medicine, National Taiwan University, Taipei 10617, Taiwan; wenwenfen@gmail.com; 4Department of Surgery, Hualien Tzu Chi Hospital, Buddhist Tzu Chi Medical Foundation, Hualien 97002, Taiwan

**Keywords:** B-cell activating factor (BAFF), B-cell CLL/lymphoma 10 (BCL10), angiogenesis, liver regeneration, 70% partial hepatectomy

## Abstract

B-cell activating factor (BAFF) is found to be associated with the histological severity of nonalcoholic steatohepatitis (NASH). BAFF was also found to have a protective role in hepatic steatosis via down regulating the expression of steatogenesis genes and enhancing steatosis in hepatocytes through BAFF-R. However, the roles of BAFF during liver regeneration are not well defined. In this study, C57/B6 mice with 70% partial hepatectomy were used as a liver regeneration model. BAFF expression was determined by enzyme immunoassay, and anti-BAFF-neutralizing antibodies were administered to confirm the effects of BAFF on liver regeneration. Western blotting, immunohistochemistry, and florescence staining determined the expression of B-cell CCL/lymphoma 10 (BCL10). The angiogenesis promoting capability was evaluated after the transfection of cells with siRNA targeting BCL10 expression, and the role of NF-κB was assessed. The results revealed that the BAFF and BCL10 levels were upregulated after partial hepatectomy. Treatment with anti-BAFF-neutralizing antibodies caused death in mice that were subjected to 70% partial hepatectomy within 72 h. In vitro, recombinant BAFF protein did not enhance hepatocyte proliferation; however, transfection with BCL10 siRNA arrested hepatocytes at the G2/M phase. Interestingly, conditioned medium from BAFF-treated hepatocytes enhanced angiogenesis and endothelial cell proliferation. Moreover, Matrix metalloproteinase-9 (MMP-9), Fibroblast growth factor 4 (FGF4), and Interleukin-8 (IL-8) proteins were upregulated by BAFF through BCL10/NF-κB signaling. In mice that were treated with anti-BAFF-neutralizing antibodies, the microvessel density (MVD) of the remaining liver tissues and liver regeneration were both reduced. Taken together, our study demonstrated that an increased expression of BAFF and activation of BCL10/NF-κB signaling were involved in hepatocyte-driven angiogenesis and survival during liver regeneration.

## 1. Introduction

The liver is a unique organ with the ability to completely regenerate to the original size after massive loss. Seventy percent partial hepatectomy is the standard model for studying normal liver regeneration. [1] The orchestrated complex process involves a cytokine, growth factor, and metabolic network [2], among which the cytokine network is initiated through the binding of tumor necrosis factor (TNF) to TNF receptor 1 (TNFR1), which leads to the activation of nuclear factor (NF)-κB in nonparenchymal cells, the production of interleukin (IL)-6, and the activation of signal transducer and activator of transcription 3 (STAT3) in hepatocytes [2]. In fact, TNFR1^–/–^ mice showed delayed regeneration attributed to inefficient activation of NF-κB [3].

Restoration of liver mass involves the proliferation of hepatocytes and nonparenchymal cells. Angiogenesis is essential for successful liver regeneration. [4,5,6,7] Mutual growth-regulatory signaling interactions between hepatocytes and endothelial cells during liver regeneration after 70% partial hepatectomy involve vascular endothelial growth factor (VEGF), IL-6, transforming growth factor (TGF) α, fibroblast growth factor (FGF) 1, and hepatocyte growth factor (HGF) [5]. Thus, interactions of multiple pathways during liver regeneration are evolutionarily preserved for host survival.

B-cell activating factor (BAFF) is a type II transmembrane glycoprotein that belongs to the TNF super family [8]. BAFF-mediated signaling involves B-cell CCL/lymphoma 10 (BCL10) nuclear translocation, changes in phospho-AKT levels, and NF-κB transactivity [9,10]. BCL10-related signaling controls the growth of cervical cancer cells via NF-κB-dependent cyclin D1 regulation [11]. Three BAFF receptors (BAFF-R, transmembrane activator and CAML interactor (TACI), and B-cell maturation antigen (BCMA)), which can specifically activate B lymphocytes and promote their proliferation, have been identified [12].

The clinical significance of BAFF is found with serum BAFF levels in patients with nonalcoholic steatohepatitis (NASH) patients had higher levels than patients with simple steatosis; meanwhile, the histological findings also demonstrated that higher BAFF levels were associated with the presence of hepatocyte ballooning and advanced fibrosis [13]. Furthermore, BAFF was also found to have a protective role in hepatic steatosis via down regulating the expression of steatogenesis genes and enhancing steatosis in hepatocytes through BAFF-R [14]. However, the role of BAFF in liver regeneration has not yet been fully elucidated.

Accordingly, we hypothesize that BAFF might play a role in liver regeneration after 70% partial hepatectomy. In this study, we investigated the role of BAFF in liver regeneration. Our findings provided important insights into the role of BAFF in angiogenesis and liver regeneration. 

## 2. Results

### 2.1. Relationship between Increased Expression of BAFF and BCL10 and Survival during Liver Regeneration 

BAFF levels in liver tissue were significantly increased at 6 h after partial hepatectomy and peaked at 16 h. However, these changes in BAFF levels were not detected in the serum (Figure 1A). As BCL10 is involved in BAFF-mediated signaling, we then evaluated the status of BCL10 in liver tissues. Western blotting (Figure 1B) and corresponding quantitative results revealed that BCL10 expression was significantly enhanced 24 h after partial hepatectomy in regenerative remnant liver tissues. Moreover, IHC confirmed these findings and showed that BCL10 expression was elevated in regenerative remnant liver tissues (Figure 1C).

Mice were intraperitoneally injected with 100 μg anti-mouse BAFF-neutralizing antibodies after 70% partial hepatectomy to clarify the role of BAFF expression in liver regeneration. We found that treatment with anti-BAFF-neutralizing antibodies, but not control IgG, caused death in mice that were subjected to 70% partial hepatectomy within 72 h (Figure 1D). These results demonstrated that BAFF was essential for survival during liver regeneration.

### 2.2. BAFF/BCL10 Signaling Plays an Important Role in Hepatocyte Proliferation

The role of BAFF/BCL10 signaling in hepatocytes is not well defined. Therefore, we used the normal human embryonic liver cell line CL-48 cells [15] to evaluate the BAFF/BCL10 signaling pathway. We first determined the BAFF receptor expression in the CL-48 cells (Figure 2A) via comparing with PBMC, which was used as BAFF receptor positive expression control. The results demonstrated that the BAFF receptor is expressed in CL-48 hepatocytes. CL-48 cells were treated with recombinant BAFF, and BCL10 expression was determined by immunofluorescence staining. BCL10 was visibly upregulated and localized to the hepatocyte nuclei (Figure 2B). BCL10 siRNA was used to knockdown BCL10 to further clarify the role of BAFF/BCL10 signaling (Figure 2C). First, we determined the effects of BAFF and BCL10 on hepatocyte growth. The results demonstrated that BAFF did not enhance the growth of hepatocytes. However, transfection with BCL10 siRNA significantly inhibited the growth of hepatocytes (Figure 2D). Moreover, flow cytometric analysis showed that transfection with BCL10 siRNA caused a significant arrest of cells in the G_2_/M phase of the cell cycles (Figure 2E).

### 2.3. BAFF Promoted Hepatocyte-Mediated Angiogenesis

The liver is a vessel-rich organ; the capability of hepatocytes to undergo angiogenesis is critical for the maintenance of liver function. TNF-α, a potent inhibitor of endothelial cell growth in vitro, is angiogenic in vivo. Therefore, we next evaluated the role of BAFF in angiogenesis in hepatocytes while using in vitro angiogenesis assays with HUVECs. The results demonstrated that, when compared with the control group, CM from BAFF-stimulated hepatocytes induced gap formation and permeability changes (Figure 3A), migration (Figure 3B), tube formation on matrix gel (Figure 3C), and proliferation (Figure 3D) of endothelial cells. However, none of these effects were observed when CM from BCL10 siRNA-transfected hepatocytes (with or without BAFF treatment) was applied. These results demonstrated that BAFF stimulation might promote hepatocyte-driven angiogenesis.

### 2.4. BAFF/BCL10/NF-κB Signaling in Hepatocytes Enhanced the Expression of Angiogenesis-Related Factors

Various factors promote angiogenesis. In this study, we used a commercial protein array to identify BAFF/BCL10-activated angiogenesis-related factors in CM from hepatocytes. The results of the array (Figure 4A,B) revealed that MMP-9, FGF4, and IL-8 were upregulated in CM from BAFF-stimulated hepatocytes when compared with those in the control cells and cells transfected with bcl10 siRNA following BAFF stimulation. Thus, BAFF/BCL10 activated the angiogenesis-related factors MMP-9, FGF4, and IL-8 in hepatocytes, resulting in increased levels in CM.

BAFF/BCL10 signaling has been found to be involved in NF-κB activation, and MMP-9, FGF4, and IL-8 are regulated by NF-κB. Therefore, we further investigated the roles of BAFF/BCL10/NF-κB signaling in hepatocytes while using NF-κB binding site-driven luciferase assays; the results revealed that BAFF significantly increased NF-κB activity and that the induction of NF-κB was significantly reduced by transfection with bcl10 siRNA or the NF-κB chemical inhibitor BAY117082 (Figure 4C). The results demonstrated the importance of BAFF/BCL10/NF-κB signaling in hepatocytes. 

Next, we determined the effects of BAFF on MMP-9, FGF4, and IL-8 induction and the role of BAFF/BCL10/NF-κB signaling in determining MMP-9, FGF4, and IL-8 protein (Figure 4D) and mRNA expression (Figure 4E) in hepatocytes. The results demonstrated that BAFF significantly enhanced MMP-9, FGF4, and IL-8 protein and mRNA expression, whereas transfection with bcl10 siRNA and treatment with BAY117082 significantly blocked these changes in MMP-9, FGF4, and IL-8 expression. Accordingly, we concluded that BAFF enhanced angiogenesis in hepatocytes by promoting the expression of angiogenesis-related factors through a pathway involving BCL10 and NF-κB signaling.

### 2.5. Downregulation of BAFF Reduced Angiogenesis and Hepatocyte Proliferation in a Liver Regeneration Model

Based on our in vitro study, we clarified the role of BAFF during liver regeneration by the administration of anti-BAFF-neutralizing antibodies to liver regeneration model mice that were subjected to 70% partial hepatectomy. After 48 h, the mice were sacrificed and the remaining liver tissues were dissected to identify the vessels positive for CD31 (Figure 5A; upper panel), an immunohistochemical marker of endothelial cells, and for Ki67 (Figure 5A; lower panel), a marker of cell proliferation. The quantitative result revealed that, when compared to the control group, mice that were treated with anti-BAFF-neutralizing antibodies showed reduced microvessel density (MVD), which corroborates the CD31-positive staining profiles (Figure 5B). Moreover, when compared to the control group, mice that were administered with anti-BAFF-neutralizing antibodies showed reduced Ki67 staining and liver regeneration (Figure 5C). Furthermore, as mentioned above, the level of MMP-9, an important angiogenic factor regulated by BAFF in the in vitro hepatocyte model, was determined in the remaining liver tissues. The results revealed that the administration of anti-BAFF-neutralizing antibodies significantly reduced the MMP-9 levels in liver tissue (Figure 5D). The results demonstrated that BAFF expression was involved in angiogenesis and hepatocyte proliferation in vivo. Please confirm that this is correct.

In summary, our findings suggested that BAFF expression was involved in hepatocyte-driven angiogenesis in liver regeneration, as illustrated in Figure 6.

## 3. Discussion

BAFF binds to the BAFF receptor to control B-cell differentiation into plasma cells and promote B-cell survival by activating the NF-κB and phosphoinositol 3-kinase/AKT pathways [16]. BAFF is induced by the nongenomic signaling of dioxin in the livers of C57BL/6 mice and HepG2 human hepatoma cells, and BAFF expression significantly contributes to early stress response reaction [17]. In this study, we observed that BAFF was upregulated in the remaining liver tissues after partial hepatectomy, which suggests that BAFF expression might be an important signal for liver regeneration. Although serum BAFF levels were not significantly increased, the local effects of BAFF in the liver tissue may be sufficient to affect regeneration signaling.

The survival of mature resting B cells in the periphery depends on signaling from the B-cell receptor and BAFF of the TNF receptor family. BCL10 promotes NF-κB activity, which contributes to B-cell survival through activation of the inhibitor of NF-κB kinase complex via Carma1 and mucosa-associated lymphoid tissue lymphoma translocation gene 1 and increases the expression of survival genes by directly modifying the chromatin of NF-κB target gene promoters. [18,19] Increased NF-κB activity and elevated cyclin D1 expression are critical for hepatocyte proliferation. [20] In this study, the CL-48 cells were treated with recombinant BAFF, and BAFF/BCL10 signaling was assessed by the detection of BCL10 nuclear translocation. However, there were no significant changes in CL-48 cell proliferation in response to BAFF treatment. In contrast, when CL-48 cells were transfected with BCL10 siRNA, cell growth retardation and cell cycle arrest at G_2_/M phase were observed. Thus, BCL10 might trigger cell proliferation signaling without BAFF activation. Consistent with this, BCL10 controls the growth of cervical cancer cells via NF-κB-dependent cyclin D1 regulation in cervical cancer cells [11].

Serum BAFF levels have been shown to correlate with parameters of disease activity, such as bone marrow microvascular density and proliferating cell nuclear antigen expression, in patients with myeloma [21]. The inhibition of BAFF expression might have therapeutic applications because of its effects on angiogenesis in human multiple myeloma [22]. In MH7A synovial cells, TNF-α-induced BAFF expression controls VEGF-mediated angiogenesis by increasing the transcription and activity of VEGF [23]. In this study, we observed that CM from BAFF-stimulated CL-48 cells promoted angiogenesis, a process that is essential for liver regeneration. sFlt-1, which is a soluble receptor for VEGF, acts as a dominant-negative receptor and it has been shown to suppress sinusoidal endothelial cell growth and reduce remnant hepatic weight [24]. Immune cells also have important roles in liver regeneration. Indeed, in mice lacking the monocyte adhesion molecule CD11b, partial hepatectomy resulted in severe reduction in angiogenesis and the development of unstable, leaky vessels, eventually producing an aberrant hepatic vascular network and Küpffer cell distribution [25].

In this study, by using an angiogenesis-related protein array, we identified the BAFF-regulated angiogenesis factors MMP-9, FGF4, and IL-8. We also confirmed the transcriptional and translational regulation of MMP-9 and IL-8 through BAFF/BCL10/NF-kB signaling in hepatocytes. Importantly, the angiogenic role of MMP-9 was first identified in homozygous mice with a null mutation in the gene encoding MMP-9/gelatinase B, which revealed an abnormal pattern of skeletal growth plate vascularization and ossification. Additionally, growth plates from gelatinase B-null mice in culture showed a delayed release of angiogenic activators, demonstrating a role of MMP-9 in angiogenesis control [26]. A stress-induced increase in MMP-9 expression are critical for recruitment of human CD34+ progenitors with hematopoietic and/or hepatic-like potential to the livers of NOD/SCID mice [27]. In MMP-9-knockout mice, a delayed hepatic regenerative response after partial hepatectomy was observed [28], highlighting the importance of MMP-9 in liver regeneration. IL-8 is a cytokine that acts as a chemoattractant for lymphocytes and neutrophils. The role of IL-8 in angiogenesis was first demonstrated in a rabbit corneal pocket model, where IL-8 induced neovascularization [29]. The angiogenic role of IL-8 is also evident by its ability to induce proliferation and chemotaxis in HUVECs [30]. In humans, after liver surgery, IL-8 is produced in the remaining liver [31]. Notably, NF-κB regulates the expression of both MMP-9 and IL-8 [32,33]. Consistent with this, in the current study, we also confirmed the role of NF-κB in BAFF/BCL10 signaling by promoter assays and chemical inhibition.

The results demonstrated that the BAFF/BCL10/NF-κB signaling pathway was active in hepatocytes and it was involved in modulating the expression of angiogenesis-related factors. We also found that the inhibition of BAFF expression reduced angiogenesis and hepatocyte proliferation in a liver regeneration model. Thus, these results again confirmed the role of BAFF in liver regeneration and suggested that drugs targeting the BAFF/BCL10/NF-κB signaling pathway should be carefully used in patients with liver regeneration-related conditions.

Importantly, one limitation of this study was the lack of human tissue validation in regenerating liver based on the critical roles of BAFF/BCL10 in angiogenesis in animal models. Thus, further studies are required to investigate the potential interactions between BAFF/BCL10 and IL-6 signaling pathways, both of which involve activation of NF-κB.

In this study, although the HUVECs model demonstrated the role of BAFF in angiogenesis in hepatocytes, HUVECs are quite different than liver sinusoidal endothelial cells (LSECs). LSECs represent a permeable barrier which representing the interface between blood cells on the one side and hepatocytes and hepatic stellate cells on the other side are highly specialized endothelial cells. Furthermore, the absence of diaphragm and lack of basement membrane make LSECs the most permeable endothelial cells of the mammalian body [34]. It is worthy to investigate the BAFF promoted hepatocyte-driven angiogenesis in LSECs.

In conclusion, the increased expression of BAFF and activation of BCL10/NF-κB signaling were critically involved in hepatocyte-driven angiogenesis and survival during liver regeneration.

## 4. Materials and Methods

### 4.1. Animals and Grouping

Male C57/B6 mice weighing approximately 25 g were used in this study. The mice were sacrificed at 0, 6, 16, 24, and 48 h after hepatectomy. All animal use protocols were reviewed and approved by the Institutional Animal Care and Use Committee (IACUC) of National Taiwan University College of Medicine and College (IACUC Approval No: 20140146).

### 4.2. Surgical Procedures

All the mice were subjected to inhalational anesthesia by isoflurane (2-chloro-2-[difluoromethoxy]-1,1,1-trifluoroethane). A midline laparotomy was performed. Partial hepatectomy was then carried out while using aseptic extirpation of the median and left lateral lobes (around 70%). The removed liver sample was immediately weighed. Laparotomy with the manipulation of the liver was carried out in sham-operated mice.

### 4.3. Tissue Processing

The animals were anesthetized with isoflurane, blood samples were collected via cardiac puncture for serum isolation, and the remaining livers were immediately removed. Part of the liver was fixed in 10% neutral-buffered formalin, embedded in paraffin, and sectioned for immunohistochemistry (IHC). The other liver tissue was used fresh for total protein extraction for BAFF determination by enzyme-linked immunosorbent assays (ELISAs) or for BCL10 determination by western blot analysis. Proteins were extracted while using Cell Lysis (Total Protein Extraction) buffer (Thermo Fisher Scientific, Waltham, MA, USA).

### 4.4. Immunohistochemical Staining and Quantification

Slides were rehydrated in phosphate-buffered saline (PBS) for 15 min., and endogenous peroxidases were inhibited by treatment with 3% H_2_O_2_/methanol for 10 min. at room temperature (25 °C). For blocking, 5% nonfat milk/PBS was used for 30 min. at room temperature. Slides were incubated with anti-BCL10(sc-5273, dilution used 1:50; Santa Cruz Biotechnology, Dallas, TX, USA), Ki67(sc-7846, dilution used 1:200; Santa Cruz Biotechnology), and CD31 (sc-376764, dilution used 1:200; Santa Cruz Biotechnology) antibodies for 16 h at 4 °C and then with peroxidase-conjugated secondary antibodies for 1 h at room temperature. The slides were then developed by immersion in 0.06% 3,3′-diaminobenzidine tetrahydrochloride (DAB; DAKO, Glostrup, Denmark), followed by counterstaining with Gill’s hematoxylin V.

IHC reactions for CD31 and Ki67 were imaged at low magnification (×40) and CD31 or Ki67-positive cells were counted in 10 representative high power fields (×400). Single immunoreactive endothelial cells, or endothelial cell clusters that were separate from other microvessels, were counted as individual microvessels. The mean visual microvessel density for CD31 was calculated. Ki67-positive cells were only counted in hepatocytes, which are with large cells with cuboidal morphology.

### 4.5. BAFF, Matrix Metalloproteinase (MMP)-9, Fibroblast Growth Factor 4 (FGF4), and Interleukin-8 (IL-8) Determination

Mouse BAFF levels in serum or tissue lysates, mouse matrix metalloproteinase-9 (MMP-9) levels in tissue lysates, and human MMP-9, FGF4, and IL-8 levels in conditioned medium were determined while using enzyme immunoassay (EIA) kits (R&D Systems, Minneapolis, MN, USA).

### 4.6. Culture of CL-48 Hepatocytes and Human Umbilical Vein Endothelial Cells (HUVECs)

Human normal CL-48 hepatocytes were obtained from (Manassas, VA, USA). The cells were maintained in Dulbecco’s modified Eagle’s medium (DMEM) supplemented with nonessential amino acids, l-glutamine, a 2× vitamin solution (Life Technologies Inc., Grand Island, NY, USA), sodium pyruvate, 10% fetal bovine serum, penicillin, and streptomycin (Flow Labs, Rockville, MD, USA). HUVECs were obtained from Cell Applications (San Diego, CA, USA) and then maintained in endothelial cell growth medium (Cell Applications). HUVECs were used at no more than passage 5. All of the cells were cultured at 37 °C in a humidified atmosphere of 5% CO_2_ and 95% air.

### 4.7. BAFF Protein and Chemical Inhibitors

Recombinant human BAFF protein and anti-mouse BAFF neutralizing antibody were obtained from R&D Systems. BAY117082 was purchased from Sigma (St. Louis, MO, USA).

### 4.8. Preparation of Conditioned Medium (CM)

The CL-48 cells were washed with PBS twice and cultured in 5 mL serum-free DMEM for 24 h before stimulated with recombinant human BAFF protein for 1 h. The CL-48 cells were then washed with PBS twice and cultured in 5 mL serum-free M199 medium for 24 h. CM was then collected and clarified by centrifugation (4 °C, 1000× *g*, 5 min.) to remove cell debris. A solution of 25 mM HEPES buffer (pH 7.4), 1 mg/mL leupeptin, 1 mM phenylmethylsulfonyl fluoride, 1 mM ethylenediaminetetraacetic acid (EDTA), 0.02% NaN_3_, and 0.1% bovine serum albumin (Sigma) was added to the CM. The CM was finally frozen and stored at −70 °C until use.

### 4.9. Cell Growth Determination

The CL-48 cells were plated in six-well cell culture plates at 20,000 cells/well in 2 mL culture medium containing fetal bovine serum. The cells were treated as indicated and harvested by suspension in 0.025% trypsin containing 0.02% EDTA. Cell counts were performed in triplicate while using a hemocytometer with trypan blue exclusion to identify the viable cells. Growth curves were generated.

Cell cycle analysis of CL-48 cells was carried out by quantifying the DNA content with propidium iodide staining and using a FACScan instrument with CellQuest software (Becton Dickinson).

### 4.10. Filamentous Actin (F-actin) Fluorescence Staining

HUVECs were cultured on cover slides and, after reaching confluence, the HUVECs were treated with CM. After 1 h, HUVECs were washed with serum-free M199 medium, fixed with 3.7% paraformaldehyde for 20 min, and permeabilized with 0.1% Triton-X-100. Fluorescence isothiocyanate-conjugated phalloidin (Invitrogen, Carlsbad, CA, USA), diluted in PBS (2 U/mL), was then applied to the specimens in the dark for 1 h. The specimens were mounted with 10% glycerol and images were acquired using a fluorescence microscope (Nikon, Tokyo, Japan).

### 4.11. HUVEC Monolayer Permeability Assays

HUVECs were cultured in Transwell chambers (0.4 μm pore polycarbonate filters; Costar, Cambridge, MA, USA). After reaching confluence, the medium was replaced with the CM (0.3 mL in the upper chamber and 1 mL in the lower chamber). Horseradish peroxidase molecule (Sigma-Aldrich, Saint Louis, MO, USA) was added to the upper compartment at a concentration of 0.2 µM. After incubation for 1 h, the medium in the lower compartment was assayed for enzymatic activity while using a photometric guaiacol substrate assay (Sigma-Aldrich).

### 4.12. HUVEC Migration Assays

Confluent HUVECs were grown in 24-well plates. To rule out the confounding influence of cell proliferation, the HUVECs were treated with 10 μg/mL mitomycin C (Sigma-Aldrich) for 2 h prior to the migration assay. A small linear scratch was created in the confluent monolayer by gently scraping with a sterile cell scrapper. The cells were extensively rinsed with M199 medium to remove cellular debris and then incubated with CM. Six hours later, images of the migrated cells were digitally photographed. The degree of wound closure was determined while using ImageJ^®^ program to measure the percent closure of the wounded area within the captured images.

### 4.13. HUVEC Tube Formation Assays

HUVECs (2 × 10^4^ cells/well in 96-well plates) were plated onto a thin coating of Matrigel (0.24 mg/cm^2^) with CM. After 6 h, each well was digitally photographed through phase contrast microscopy. Tubes are defined that develop contain a lumen encircled by endothelial cells that are joined together via junctional complexes and the number of intact tubes was counted per high power field.

### 4.14. HUVEC Proliferation Tests

The cells were plated in six-well cell culture plates at 1 × 10^5^ cells/well in 2 mL culture medium with CM. After 72 h of treatment at 37 °C, the cells were harvested by suspension in 0.025% trypsin with 0.02% EDTA. Cell counts were performed in triplicate while using a hemocytometer. Trypan blue exclusion assays were used to identify the viable cells. The cell number was determined and cell growth curves were generated.

### 4.15. Protein Array Analysis

A Proteome Profiler Human Angiogenesis Antibody Array (cat. #ARY007; R&D Systems) was used for the detection of angiogenesis-related factors according to the manufacturer’s instructions. Briefly, CM was first mixed with the detection antibody cocktail at room temperature for 1 h prior to being added to the array membrane. The membrane was then incubated overnight at 2–8 °C on a shaker. After washing, horseradish peroxidase-conjugated streptavidin was added to the membrane, followed by incubation for 30 min. at room temperature on a shaker. After washing, X-ray film and a chemiluminescence imaging system were used to detect and quantify the array signals.

### 4.16. RNA Interference

Small interfering RNA (siRNA) duplexes were purchased from Santa Cruz Biotechnology (Santa Cruz, CA, USA; siRNA targeting BCL10, sc-29793; and control siRNA, sc-37007). The CL-48 cells were transfected with siRNA at a concentration of 25 nM in serum-free Opti-MEM using Oligofectamine (Invitrogen, Carlsbad, CA, USA).

### 4.17. Real-Time Quantitative Reverse Transcription Polymerase Chain Reaction (RT-PCR)

We quantified mRNA expression under various conditions while using a fluorescence quantitative real-time PCR detection system (Light Cycler DNA master SYBR Green I; Roche Molecular Biochemicals, Indianapolis, IN, USA). The primer pairs were as follows: human BAFF-R: 5′-AGACAAGGACGCCCCAGAGCCC-3′ and 5′-GTGGGGTGGTTCCTGGGTCTTC-3′; hMMP-9: 5′-CACTGTCCACCCCTCAGAGC-3′ and 5′-GCCACTTGTCGGCGATAAGG-3′; *IL-8*: 5′-TTTCTGCAGCTCTCTGTGAGG-3′ and 5′-CTGCTGTTGTTGTTGCTTCTC-3′; FGF4: 5′-GACTACCTGCTGGGCATCAA-3′ and 5′-TGCACTCATCGGTGAAGAAG-3′; glyceraldehyde-3-phosphate dehydrogenase (*GAPDH*), 5′-GGGAAGGTGAAGGTCGG-3′ and 5′-TGGACTCCACGACGTACTCAG-3′. Amplification was followed by melting curve analysis to verify the authenticity of the amplicon. The amounts of BAFFR, MMP-9, FGF4, and IL-8 mRNAs were normalized to that of GAPDH mRNA and they are presented in arbitrary units, with 1 U corresponding to the value in cells that were treated with the vehicle control.

### 4.18. NF-κB Promoter Reporter Assays

Transfection with NF-κB binding site-driven luciferase plasmid (BD Bioscience) into CL-48 cells was performed while using Transfast transfection reagent (Promega, Madison, WI, USA). At 24 h after transfection, the cells were serum starved for 24 h and then treated as indicated.

### 4.19. Western Blotting

Protein concentrations in nuclear extracts from liver tissues or CL-48 cell lysates were quantified while using Bio-Rad protein assays (Hercules, CA, USA). Samples (10–50 μg protein) were separated by sodium dodecyl sulfate polyacrylamide gel electrophoresis, transferred onto polyvinylidene difluoride membranes, and immunoblotted with anti-BCL10 (sc-5273, dilution used 1:200; Santa Cruz Biotechnology) and anti-beta-Actin (sc-47778, dilution used 1:1000; Santa Cruz Biotechnology) antibodies. The entire western blots with marker were demonstrated in Appendix A. The signals were detected by a Digital imaging system (Bio Pioneer Tech Co., New Taipei City, Taiwan). Meanwhile, the relative intensity of thesignals was analyzed with the ImageJ^®^ program. Is the capitalization necessary? Additional examples will be highlighted below.

### 4.20. Statistical Analysis

Data are expressed as mean ± standard deviations (SDs), one- or two-way analysis of variance (ANOVA) with Turkey post hoc test was used to analyze the data in multiple groups. The Student’s *t*-test was used to evaluate statistically significant differences between the groups. All statistical analyses were performed while using SPSS for Windows, version 18.0 (SPSS Inc., Chicago, IL, USA). A *p* value of less than 0.05 was considered to indicate statistical significance.

## Figures and Tables

**Figure 1 ijms-20-05022-f001:**
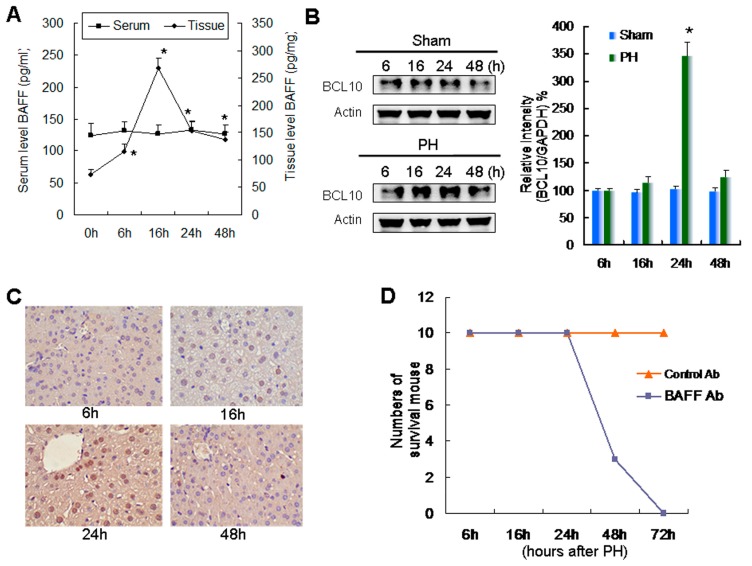
B-cell activating factor (BAFF) and B-cell CCL/lymphoma 10 (BCL10) expression in 70% partial hepatectomy-induced liver regeneration in mice. (**A**) Serum and tissue levels of BAFF were estimated by EIA at the indicated times after 70% partial hepatectomy. Data are presented as means ± SDs, and comparisons were made between each time and time zero. *n* = 6. * *p* < 0.05, by two-way ANOVA with Tukey’s post hoc test. (**B**) Left panel, expression levels of BCL10 at different times in liver tissues from control or 70% partial hepatectomy (PH) groups were determined by western blotting; Acin was used as loading control. Right panel, the quantitative results of BCL10 western blotting. Data are presented as the relative intensity (BCL10/Actin) ± SD. Comparisons were made between the control and PH groups. *n* = 6. * *p* < 0.05, by Student’s *t*-test. (**C**) Tissue BCL10 staining (brown color) of remnant liver tissues. Magnification, 400×. (**D**) Mice were intraperitoneally injected with 100 μg control IgG (Control Ab) or anti-mouse BAFF-neutralizing antibodies (BAFF Ab) at the time of 70% PH. The number of surviving mice was calculated at different times after 70% PH. *n* = 10 per group.

**Figure 2 ijms-20-05022-f002:**
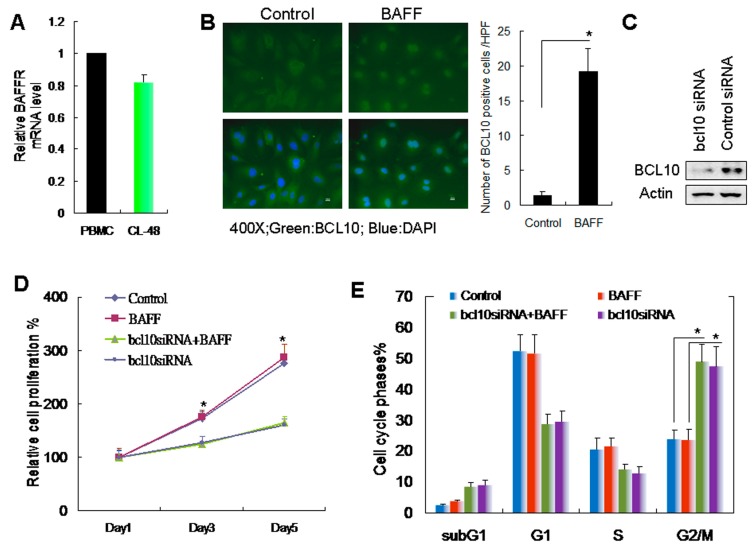
BAFF/BCL10 signaling in hepatocye cell proliferation. (**A**) The expression of BAFFR mRNA in human CL-48 hepatocytes was determined by q-Reverse Transcription Polymerase Chain Reaction (q-RT-PCR); commercialized human peripheral blood mononuclear cells (PBMC) cDNA was used as the positive control. (**B**) Left panel, human CL-48 hepatocytes were treated without (control) or with BAFF (1 ng/mL) for 1 h, and the expression of BCL10 was determined by immunofluorescence staining; BCL10 was identified as a green signal, and the nucleus was stained with DAPI (blue). Magnification, 400×. Right panel, the number of BCL10 positive cells was counted under high power field (HPF). *n* = 6. * *p* < 0.05, by Student’s *t*-test. (**C**) CL-48 cells were treated with control or with BCL10 siRNA for 24 h; the expression of BCL10 was determined by western blotting. Actin was used as the loading control. (**D**) CL-48 cells were treated with control or BCL10 siRNA for 24 h prior treatment with BAFF (1 ng/mL). At different time points, relative cell proliferation was determined by the Trypan blue exclusion assay. The data are shown as mean ± SD of three independent experiments. Comparisons were made between the BAFF and bcl10siRNA + BAFF groups. * *p* < 0.05, by Student’s *t*-test. (**E**) CL-48 cells were treated as indicated with BCL10 siRNA or BAFF, as described in (**D**). On day 3, the cell cycle phases were determined by propidium iodide staining and FACScan analysis. Populations of cells in the sub-G_1_, G_1_, S, and G_2_/M phases were analyzed and quantified while using the Cell Quest software, and the data represent the means  ± SDs of three independent experiments. Between-group comparisons were performed as indicated. * *p* < 0.05, by Student’s *t*-test.

**Figure 3 ijms-20-05022-f003:**
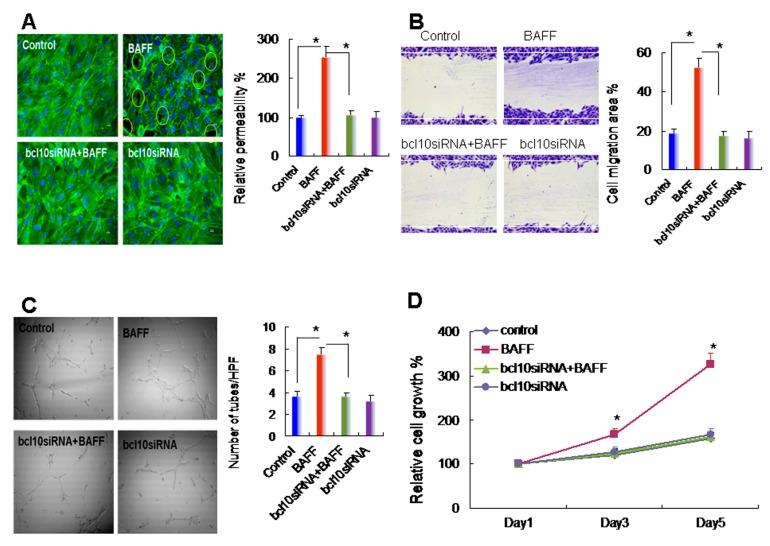
Effects of conditioned medium derived from BAFF-stimulated hepatocytes on angiogenesis. Conditioned medium from BAFF-stimulated CL-48 cells was used for angiogenic function assays. Four experimental conditions were used: control, the conditioned medium was collected from hepatocytes without any treatment; BAFF group, conditioned medium was from BAFF (1 ng/mL)-stimulated hepatocytes; BCL10siRNA+BAFF group, conditioned medium was from BCL10 siRNA-transfected hepatocytes following BAFF stimulation; and BCL10siRNA group, conditioned medium was from BCL10 siRNA-transfected hepatocytes. All conditioned medium was collected after 24 h culture. (**A**) **Left panel**, confluent and Human Umbilical Vein Endothelial Cell (HUVEC) monolayers were treated with conditioned medium for 1 h, and gap formation was determined by phalloidin staining. **Right panel**, conditioned medium from the indicated conditions was tested with HUVEC monolayers for 1 h using permeability assays. Data are shown as the relative permeability percentages, with the control conditioned medium in lane 1 defined as 100%. Comparisons are shown between the indicated groups. * *p* < 0.05. *n* = 5, by one-way ANOVA with Tukey’s post hoc test. (**B**) Left panel, HUVECs were treated with conditioned medium for 6 h for migration assays, and representative images of migrated HUVECs in each group are shown. Right panel, quantitative results of migrated HUVECs was used to calculate the cell migration area. Comparisons are shown between the indicated groups. * *p* < 0.05. *n* = 5, by one-way ANOVA with Tukey’s post hoc test. (**C**) **Left panel**, HUVECs were treated with conditioned medium for 6 h for capillary tube formation assays, and representative images of tube formation in each group are shown. **Right panel**, quantitative results of tube formation was calculated under high-power fields. Comparisons are shown between the indicated groups. * *p* < 0.05. *n* = 5, by one-way ANOVA with Tukey’s post hoc test. (**D**) HUVECs were treated with conditioned medium for 1–5 days for cell growth determination by Trypan blue exclusion assays. Data are the relative cell growth percentages for the indicated conditions, with the control group in lane 1 defined as 100%. *n* = 5. Comparisons were made between the BAFF and bcl10siRNA+BAFF groups. * *p* < 0.05, by Student’s *t*-test.

**Figure 4 ijms-20-05022-f004:**
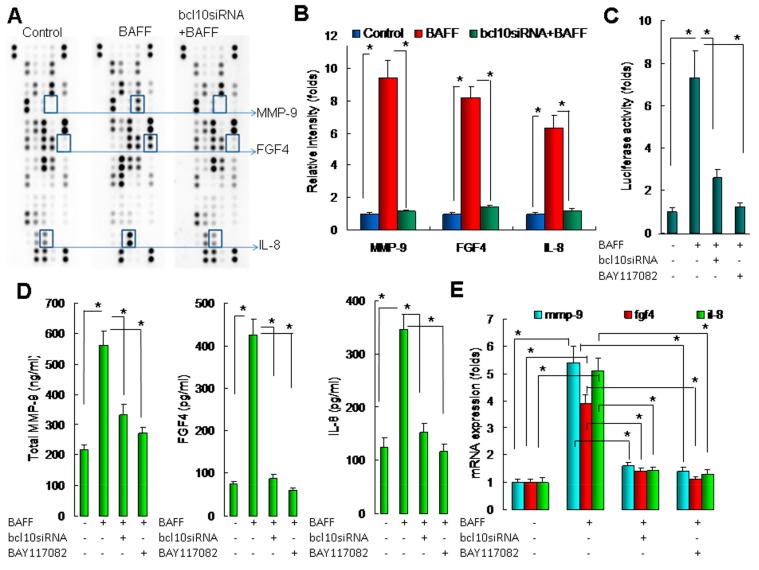
BAFF stimulated Matrix metalloproteinase-9 (MMP-9), Fibroblast growth factor 4 (FGF4), and Interleukin-8 (IL-8) expression in hepatocytes. (**A**) Conditioned medium from the three experimental conditions was applied to protein array analysis for identification of angiogenesis-related factors. Four experimental conditions were used: control, the conditioned medium was collected from hepatocytes without any treatment; BAFF group, conditioned medium was from BAFF (1 ng/mL)-stimulated hepatocytes; bcl10siRNA+BAFF group, conditioned medium was from bcl10 siRNA-transfected hepatocytes following BAFF stimulation; and bcl10siRNA group, conditioned medium was from bcl10 siRNA-transfected hepatocytes. All conditioned medium was collected after 24 h of culture. Data shown are representative images of three independent experiments. Significantly altered protein spots are indicated. (**B**) Quantitative results for MMP-9, FGF4, and IL-8. Results are the relative intensity of spots from the angiogenesis protein arrays. *n* = 3. * *P* < 0.05, by Student’s *t*-test. (**C**) CL-48 cells were transfected with an NF-κB binding site-driven luciferase plasmid and then transfected with bcl10 siRNA for 24 h or treated with BAY117082 (100 nM, for 1 h and then depleted the BAY117082 contained medium by twice wish with culture medium), following treatment with recombinant BAFF (1 ng/mL). After 4 h, NF-κB promoter activities were determined. Data are compared with that from lane 1. *n* = 5. * *P* < 0.05, by one-way ANOVA with Tukey’s post hoc test. (**D**) CL-48 cells were transfected with BCL10 siRNA for 24 h or treated with BAY117082 (100 nM, for 1 h and then depleted the BAY117082 contained medium by twice wish with culture medium), following treatment with recombinant BAFF (1 ng/mL). After 24 h, the protein levels of total MMP-9, FGF4 and IL-8 in the cell culture supernatants were determined by EIAs. Data are compared with that from lane 1. *n* = 5. * *P* < 0.05, by one-way ANOVA with Tukey’s post hoc test. (**E**) CL-48 cells were transfected with BCL10 siRNA for 24 h or treated with BAY117082 (100 nM, for 1 h and then depleted the BAY117082 contained medium by twice wish with culture medium), following treatment with recombinant BAFF (1 ng/mL). After 8 h, mmp-9, fgf4 and il-8 mRNA levels were determined by qRT-PCR. Data are compared with that from lane 1. *n* = 5. * *P* < 0.05, by one-way ANOVA with Tukey’s post hoc test.

**Figure 5 ijms-20-05022-f005:**
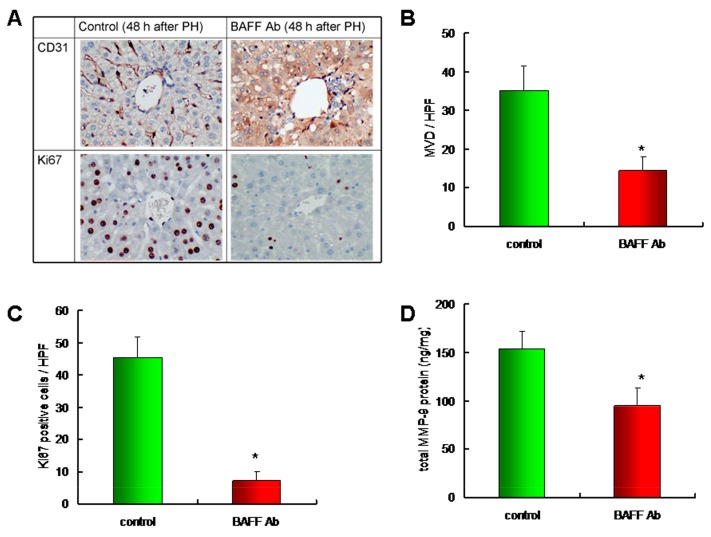
Microvessel density, cell proliferation, and MMP-9 expression in liver tissue of BAFF-neutralizing antibodies treated mice. (**A**) Mice were injected intraperitoneally with 100 μg control IgG or anti-mouse BAFF-neutralizing antibodies after 70% partial hepatectomy. After 48 h, mice were sacrificed, and the remaining liver tissues were dissected and stained for CD31 and Ki67. Magnification, 400×. (**B**) Quantitative results of microvessel density in remaining liver tissue, *n* = 10 per group. * *p* < 0.05, by Student’s *t*-test. (**C**) Quantitative results of Ki67 positive staining cells in remaining liver tissue, *n* = 10 per group. * *p* < 0.05, by Student’s *t*-test. (**D**) MMP-9 protein level in remaining liver tissue was determined by EIA, *n* = 10 per group. * *p* < 0.05, by Student’s *t*-test.

**Figure 6 ijms-20-05022-f006:**
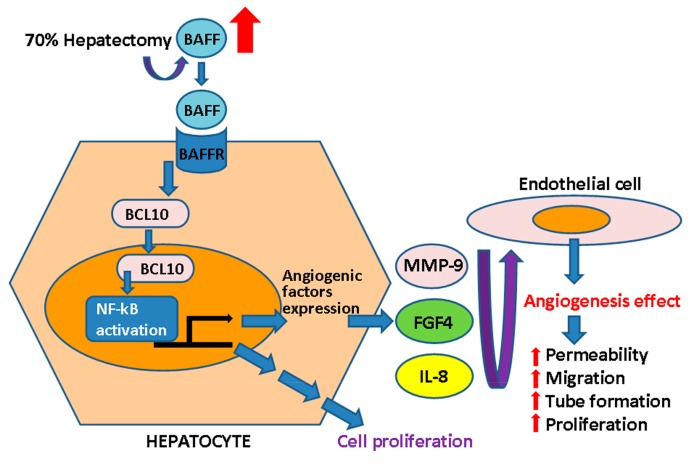
Summary of BAFF in liver regeneration. In this study, we found that BAFF was induced in the remaining liver tissue after 70% partial hepatectomy. Since BAFFR was found to be expressed in hepatocytes, we assumed the BAFF activated BCL10 nuclear translocation through BAFFR. BCL10 nuclear translocation subsequently activated NF-κB-dependent gene expression, including cell proliferation, as consistent with a previous study.^11^ Interestingly, we found that BAFF may enhance the expression of factors that promote gap formation, permeability changes, migration, tube formation on matrix gel, and endothelial cell proliferation. Moreover, we found the BAFF/BCL10/NF-κB cascade activated the angiogenesis-related factors MMP-9, FGF4, and IL-8. The results demonstrated that BAFF expression was critically involved in hepatocyte-driven angiogenesis during liver regeneration.

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
