# Peer review of "B-Cell Activating Factor Enhances Hepatocyte-Driven Angiogenesis via B-Cell CLL/Lymphoma 10/Nuclear Factor-KappaB Signaling during Liver Regeneration"

_ijms, 2019, doi:10.3390/ijms20205022_

Round 1

Reviewer 1 Report

This paper documents a new role for BAFF/BCl10/NFkb signaling in liver regeneration and hepatocyte-driven angiogenesis. The paper is generally clear and logically presented. There are only a few minor concerns which need to be addressed:

Figure 1C does not clearly shown nuclear localization as the text suggests. Please insert better figures or remove this statement from the text. Figure 2B is not sufficient. Multiple cells need to be shown and this data needs to be quantified. The legend says MTT was used for proliferation, but the methods say the cells were counted with trypan blue exclusion- please clarify. Migration assay should be done in presence of proliferation inhibitor such as mitomycin C. What happens to FGF4? In figure 4 D and E you show both MMP9 and IL8, but not FGF4. Please re-write the abstract. Begin the abstract with the full name of BAFF. This statement from the abstract is too strong based on the data “In mice treated with anti-BAFF-neutralizing antibodies, the structures of liver sinusoidal endothelial cells were lost”. The statement about NAFLD is confusing. What is the link between NAFLD /BAFF and BAFF in regeneration. Please clarify. This statement needs to be rewritten. “These results demonstrated that BAFF stimulation may promote angiogenesis in hepatocytes.” Hepatocyte-driven angiogenesis may be better. Line 265-266. HUVECs are not a good model to investigate the liver microvascular endothelial cells, as they are quite different than LSECs. This must be clearly stated in the discussion. Clarify all legends with sufficient experimental and analytical details.

Author Response

We would like to express our sincere appreciation for the valuable suggestions and comments from the reviewer!

Response to Reviewer 1 Comments

This paper documents a new role for BAFF/BCl10/NFkb signaling in liver regeneration and hepatocyte-driven angiogenesis. The paper is generally clear and logically presented. There are only a few minor concerns which need to be addressed:

Figure 1C does not clearly shown nuclear localization as the text suggests. Please insert better figures or remove this statement from the text.

Response: Thanks for the comment, we have changed the statement with IHC confirmed these findings and showed that BCL10 expression was elevated in regenerative remnant liver tissues.

Figure 2B is not sufficient. Multiple cells need to be shown and this data needs to be quantified.

Response: Thanks for the comment, we have changed the data with multiple cells, further we also quantified this data by counting the BCL10 positive cells.

The legend says MTT was used for proliferation, but the methods say the cells were counted with trypan blue exclusion- please clarify.

Response: Thanks for the comment. In this study, both CL-48 and HUVEC cells were plated in six-well cell culture plates for cell proliferation assay. The cell counts were performed in triplicate using a hemocytometer with Trypan blue exclusion assay. We have corrected the description in the figure legend of Figure 2D and Figure 3D.  

Migration assay should be done in presence of proliferation inhibitor such as mitomycin C.

Thanks for this comment; we have redone the migration assay via in presence of proliferation inhibitor mitomycin C and described in 2.12 HUVEC migration assays. The HUVECs were treated with 10 μg/mL mitomycin C for 2 hours prior to the migration assay. The results demonstrated that compared with the control group, CM from BAFF-stimulated hepatocytes induced migration of endothelial cells without the confounding influence of cell proliferation.

What happens to FGF4? In figure 4 D and E you show both MMP9 and IL8, but not FGF4. Please re-write the abstract.

Thanks for this comment; we have added the result of FGF4 in Figure 4B, 4D and 4E. Further the abstract was re-written to contain the description of FGF4. 

Begin the abstract with the full name of BAFF. This statement from the abstract is too strong based on the data “In mice treated with anti-BAFF-neutralizing antibodies, the structures of liver sinusoidal endothelial cells were lost”.

Thanks for this comment; we have corrected the abstract with the full name of BAFF. Based on our in vitro study, we clarified the role of BAFF during liver regeneration by administration of anti-BAFF-neutralizing antibodies to liver regeneration model mice subjected to 70% partial hepatectomy. After 48 h, mice were sacrificed, and the remaining liver tissues were dissected to identify the microvessel density (MVD) by comparing the performance of pan-endothelial marker CD31 (Fig. 5A; upper panel). The CD31 expression of normal LSECs is restricted to the cytoplasm rather than to the cell surface at cell-cell junctions. Quantitation of the results revealed that compared to the control group, mice treated with anti-BAFF-neutralizing antibodies showed reduced MVD, which corroborates the CD31-positive staining profiles (Fig. 5B). We have corrected the structures of liver sinusoidal endothelial cells were lost into the MVD of the remaining liver tissues was reduced. Therefore, we clarified the effect of BAFF on hepatocyte-driven angiogenesis as suggesting by reviewer in the following comment.

The statement about NAFLD is confusing. What is the link between NAFLD /BAFF and BAFF in regeneration. Please clarify. This statement needs to be rewritten.

Thanks for this comment; we have rewritten the statement about the link between NASH /BAFF and BAFF in regeneration to make the rational of this study more clearly. “The clinical significance of BAFF is found with serum BAFF levels in patients with nonalcoholic steatohepatitis (NASH) patients had higher levels than patients with simple steatosis; meanwhile, histological findings also demonstrated of higher BAFF levels were associated with the presence of hepatocyte ballooning and advanced fibrosis.13 Furthermore, BAFF was also found to have a protective role in hepatic steatosis via down regulating the expression of steatogenesis genes and enhance steatosis in hepatocytes through BAFF-R.14 However, the role of BAFF in liver regeneration has not yet been fully elucidated.”

“These results demonstrated that BAFF stimulation may promote angiogenesis in hepatocytes.” Hepatocyte-driven angiogenesis may be better.

Thanks for this comment; we have corrected the statement to “These results demonstrated that BAFF stimulation may promote hepatocyte-driven angiogenesis.” In section 3.3 BAFF promoted hepatocyte-mediated angiogenesis. Line 282. And In Figure 6 (Line 384,385 and 395)

Line 265-266. HUVECs are not a good model to investigate the liver microvascular endothelial cells, as they are quite different than LSECs. This must be clearly stated in the discussion.

Thanks for this comment; in the original manuscript Line 265-266, we described that we next evaluated the role of BAFF in angiogenesis in hepatocytes using in vitro angiogenesis assays with HUVECs. This is due to the usage of human CL-48 hepatocytes in the above study. The original thinking is to realize the effect of BAFF/BCL10 signaling does not only exist in mouse model but also present in human hepatocytes. To avoid the interference of cell species so we used HUVECs as endothelial cells model. However, as the comment by reviewer, in deed, HUVECs are not a good model to investigate the liver microvascular endothelial cells, as they are quite different than LSECs. We have added the following statement in the discussion. Line 459-465, ”In this study, although the HUVECs model demonstrated the role of BAFF in angiogenesis in hepatocytes, however, HUVECs are quite different than LSECs. LSECs represent a permeable barrier which representing the interface between blood cells on the one side and hepatocytes and hepatic stellate cells on the other side are highly specialized endothelial cells. Furthermore, the absence of diaphragm and lack of basement membrane make LSECs the most permeable endothelial cells of the mammalian body. It is worthy to investigate the BAFF promoted hepatocyte-driven angiogenesis in LSECs.

Clarify all legends with sufficient experimental and analytical details. 

Thanks for this comment; we have clarified all legends with sufficient experimental and analytical details.

Reviewer 2 Report

In this article, authors demonstrate the importance of BAFF in upregulating the pro-agiogenic properties of hepatocytes following hepatectomy. Overall the article is interesting. Here are some points that need to be addressed to improve the quality of the manuscript.

“the angiogenesis of hepatocytes” does not mean anything and has to be changed. It is rather the properties of hepatocytes to stimulate angiogenesis How were tubes defined in the tube formation assay? Provide the antibody dilutions used for the Western blot analysis Figure 1B, how many livers were analysed The methods used for figure 3D is not described in the method section. Also what groups does the statistics compare in this figure Different concentrations of BAY117082 were used in results presented in figure 4. Why did the authors change it and used fairly low dose compared to what is used in the literature. I am surprised that BAY117082 does not reduce baseline activity of NFKB (Figure 4c). How do authors explain it? What is the CD31 staining in figure 5A in the BAFF ab condition?

Author Response

We would like to express our sincere appreciation for the valuable suggestions and comments from the reviewer!

Response to Reviewer 2 Comments

In this article, authors demonstrate the importance of BAFF in upregulating the pro-agiogenic properties of hepatocytes following hepatectomy. Overall the article is interesting. Here are some points that need to be addressed to improve the quality of the manuscript.

“the angiogenesis of hepatocytes” does not mean anything and has to be changed. It is rather the properties of hepatocytes to stimulate angiogenesis

Thanks for this comment; we have corrected the statement to “These results demonstrated that BAFF stimulation may promote hepatocyte-driven angiogenesis.” And in the topic” B-cell activating factor enhances hepatocyte-driven angiogenesis via B-cell CLL/lymphoma 10/nuclear factor-kappaB signaling during liver regeneration”

How were tubes defined in the tube formation assay?

Thanks for this comment; culturing endothelial cells on a gel of basement membrane extract is a way to induce their differentiation and tube-like structures formation. We have added the defined of tubes in the 2.13 HUVEC tube formation assays: After 6 h, each well was digitally photographed through phase contrast microscopy. Tubes are defined that develop contain a lumen encircled by endothelial cells that are joined together via junctional complexes and the number of intact tubes was counted per high power field.

Provide the antibody dilutions used for the Western blot analysis Figure 1B,

Thanks for this comment; we have provided the antibody dilutions used for the Western blot analysis Figure 1B within the part of 2.19 Western blotting: Immunoblotted with anti-BCL10 (sc-5273, dilution used 1:200; Santa Cruz Biotechnology) and anti-beta-Actin (sc-47778, dilution used 1:1000; Santa Cruz Biotechnology) antibodies.

how many livers were analysed

Thanks for this comment; in this study, in different part of experiments, we used different numbers of mice to collect the livers for analyzed. In Figure 1. (A) to (C), 6 mice per group was used to collect Serum and tissue sample. In Figure 1(D), 10 mice per group were used to realize the surviving mice. In Figure 5. 10 mice per group were used to collect Serum and tissue sample.

The methods used for figure 3D is not described in the method section.

Thanks for this comment; in figure 3D, we described that HUVECs were treated with conditioned medium for 1–5 days for cell growth determination by MTT assays. Actually, we used Trypan blue exclusion assay as descriped in 2.14 HUVEC proliferation tests: Cells were plated in six-well cell culture plates at 1 × 105 cells/well in 2 mL culture medium with CM. After 72 h of treatment at 37°C, cells were harvested by suspension in 0.025% trypsin with 0.02% EDTA. Cell counts were performed in triplicate using a hemocytometer. Trypan blue exclusion assays were used to identify viable cells. The cell number was determined, and cell growth curves were generated. We have corrected the mistake.

Also what groups does the statistics compare in this figure Different concentrations of BAY117082 were used in results presented in figure 4. Why did the authors change it and used fairly low dose compared to what is used in the literature. I am surprised that BAY117082 does not reduce baseline activity of NFKB (Figure 4c). How do authors explain it?

Thanks for this comment; BAFF/BCL10 signaling has been found to be involved in NF-κB activation, and MMP-9, FGF4 and IL-8 are regulated by NF-κB. Therefore, we further investigated the roles of BAFF/BCL10/NF-κB signaling in hepatocytes using NF-κB binding site-driven luciferase assays; the results revealed that BAFF significantly increased NF-κB activity and that the induction of NF-κB was significantly reduced by transfection with bcl10 siRNA or the NF-κB chemical inhibitor BAY117082. In Figure 4C to 4E, CL-48 cells were treated with the same concentration of BAY117082 (100 nM) for 1 h prior to treatment with recombinant BAFF. We have corrected the error typing. As the valuable comment from reviewer, high dose BAY117082 is toxic to cells, in our experiences, after the treatment of BAY117082 (100 nM) for 1 h, BAY117082 was washed out by renew the culture medium twice. It is a critical step to renew BAY117082 containing medium prior the following treatment and at this situation, BAY117082 treatment does not reduce baseline activity of NFKB. We have added the detail steps in the figure legend. 

What is the CD31 staining in figure 5A in the BAFF ab condition? 

Thanks for this comment; based on our in vitro study, we clarified the role of BAFF during liver regeneration by administration of anti-BAFF-neutralizing antibodies to liver regeneration model mice subjected to 70% partial hepatectomy. After 48 h, mice were sacrificed, and the remaining liver tissues were dissected to identify the microvessel density(MVD) by comparing the performance of pan-endothelial marker CD31 (Fig. 5A; upper panel). The CD31 expression of normal LSECs is restricted to the cytoplasm rather than to the cell surface at cell-cell junctions (Am J Physiol Gastrointest Liver Physiol. 2004;287(4):G757–G763.). The CD31 staining in figure 5A in the BAFF ab condition is using antibody against CD31 (sc-376764, dilution used 1:200; Santa Cruz Biotechnology) antibodies for 16 h at 4°C and then with m-IgGκ BP-HRP secondary antibody (sc-516102, dilution used 1:1000; Santa Cruz Biotechnology) for 1 h at room temperature. Slides were then developed by immersion in 0.06% 3,3'-diaminobenzidine tetrahydrochloride (DAB; DAKO, Glostrup, Denmark), followed by counterstaining with Gill’s hematoxylin V.